

# A genomic-clinical nomogram predicting recurrence-free survival for patients diagnosed with hepatocellular carcinoma

Junjie Kong, Tao Wang, Shu Shen, Zifei Zhang, Xianwei Yang and Wentao Wang

Department of Liver Surgery & Liver Transplantation Center, West China Hospital of Sichuan University, Chengdu, Sichuan Province, China

## ABSTRACT

Liver resection surgery is the most commonly used treatment strategy for patients diagnosed with hepatocellular carcinoma (HCC). However, there is still a chance for recurrence in these patients despite the survival benefits of this procedure. This study aimed to explore recurrence-related genes (RRGs) and establish a genomic-clinical nomogram for predicting postoperative recurrence in HCC patients. A total of 123 differently expressed genes and three RRGs (*PZP*, *SPP2*, and *PRC1*) were identified from online databases via Cox regression and LASSO logistic regression analyses and a gene-based risk model containing RRGs was then established. The Harrell's concordance index (C-index), receiver operating characteristic (ROC) curves and calibration curves showed that the model performed well. Finally, a genomic-clinical nomogram incorporating the gene-based risk model, AJCC staging system, and Eastern Cooperative Oncology Group performance status was constructed to predict the 1-, 2-, and 3-year recurrence-free survival rates (RFS) for HCC patients. The C-index, ROC analysis, and decision curve analysis were good indicators of the nomogram's performance. In conclusion, we identified three reliable RRGs associated with the recurrence of cancer and constructed a nomogram that performed well in predicting RFS for HCC patients. These findings could enrich our understanding of the mechanisms for HCC recurrence, help surgeons predict patients' prognosis, and promote HCC treatment.

Corresponding author
Wentao Wang, wwt0510@163.com

## INTRODUCTION

Hepatocellular carcinoma (HCC) is the most common primary liver malignancy and a contributor to the third population of cancer-related deaths, ranking sixth among the most frequent malignancies worldwide (*Forner, Reig & Bruix, 2018*). Liver resection is the most commonly used therapeutic strategy for HCC, and accounts for an overall 5-year survival rate of ~70% (*Akoad & Pomfret, 2015*; *Orcutt & Anaya, 2018*). However, for these patients, survival is impacted by postoperative tumor recurrence (*Tabrizian et al., 2015*).

Previous studies have suggested that more than half of HCC patients would suffer from a disease relapse after a hepatectomy (*Akoad & Pomfret, 2015*). Disease recurrence was divided into two types depending on their different etiologies: those who relapsed within

2 years after surgery (early recurrence) and those whose relapse occurred more than 2 years after the operation (late recurrence) (*Sherman, 2008*). Many clinical characteristics, such as the AFP level, tumor size, vascular invasion (including microscopic and macroscopic), and HBsAg level proved to be risk factors for recurrence and several predictive models were established to predict the postoperative recurrence of HCC (*Lee, Tan & Chung, 2014*; *Qiu et al., 2017*; *Chan et al., 2018*; *He et al., 2018*). However, these studies and models only focused on the relationships between clinical traits and HCC recurrence with little focus on the crucial role of molecular data.

Advances in biomedical research and high-throughput technologies have greatly contributed to the identification of molecular biomarkers related to cancer development, recurrence, and prognosis in the past few decades (*Teufel, 2015*). A large number of mRNAs, microRNAs and other kinds of biomarkers have been identified and regarded as potential targets for cancer diagnosis and treatment (*Lee, Tan & Chung, 2014*). Biomarkers could reflect the molecular mechanisms of tumor recurrence and, as a result, the identification of reliable biomarkers could improve the accuracy in predicting cancer recurrence, thereby contributing to cancer treatment (*Gu et al., 2018*; *Long et al., 2018*). Consequently, gene signatures for recurrence and prognosis prediction are receiving more attention.

In this study we explored differently expressed genes (DEGs) between the HCC and non-tumor samples using data obtained from the Gene Expression Omnibus (GEO) and The Cancer Genome Atlas (TCGA) databases. A total of three recurrence-related genes (RRGs) were then identified and a gene-based risk model was established from the data of the TCGA dataset. Furthermore, a genomic-clinical nomogram containing the gene-based risk model and clinical characteristics was constructed to predict the 1-, 2-, and 3-year recurrence-free survival (RFS) for HCC patients. We also assessed the performance of the nomogram using the Harrell's concordance index (C-index), calibration curves, receiver operating characteristic (ROC) curves, and decision curves analysis (DCA).

## MATERIALS AND METHODS

### Dataset collection

Using "hepatocellular carcinoma" as our keyword, we searched gene expression profiles of HCC from the GEO database (https://www.ncbi.nlm.nih.gov/gds/). "*Homo sapiens*" was the term chosen for the organism parameter. The following criteria were used to screen datasets: (1) datasets were mRNA expression profiled by array; (2) datasets with more than 100 samples; (3) datasets with comparison between tumor and non-tumor samples; and (4) profiles with available expression information. Finally, to obtain more accurate results and to avoid individual heterogeneity, the top four GEO datasets with the largest sample numbers, GSE25097 (*Sung et al., 2012*), GSE76427 (*Grinchuk et al., 2018*), GSE36376 (*Lim et al., 2013*) and GSE14520 (*Roessler et al., 2010*), were selected. The details of the datasets were shown in Table S1. Meanwhile, we downloaded the RNA-seq of HCC form the TCGA database (https://cancergenome.nih.gov/), which contained 374 HCC samples and 50 non-tumor samples. The related clinical files of HCC were also obtained for further analysis. Log2 transformation was used for all of the expression data.

## Exploration of differently expressed genes and bioinformatic analysis

R Software (*R Core Team, 2019*) and the related packages "limma" and "edgR" were used to screen DEGs. The cut-off criteria with a significance of $p < 0.05$ and $|log2FC| > 1$ were used in the identification of DEGs. Overlapping analysis was performed to find DEGs among all of the four GEO datasets and the TCGA dataset and the Venn plot was completed using the online tool entitled "Calculate and draw custom Venn diagrams" (http://bioinformatics.psb.ugent.be/webtools/Venn/). In addition, the online biological tools Database for Annotation, Visualization and Integrated Discovery (DAVID, https://david.ncifcrf.gov/) and KOBAS 3.0 (http://kobas.cbi.pku.edu.cn/anno_iden.php) were used to perform Gene Ontology (GO) and Kyoto Encyclopedia of Genes and Genomes (KEGG) pathways enrichment analysis, respectively. Furthermore, the search tool for the Retrieval of Interacting Genes (STRING, https://string-db.org/cgi/input.pl) and Cytoscape Software (Version 3.6.1) were used to establish a protein–protein interaction (PPI) network complex to visualize the correlations among DEGs.

## Identification of recurrence-related genes

In this progress, three datasets (TCGA, GSE14520 and GSE76427 datasets) with detailed information about tumor recurrence were divided into two types of sets: the discovery set (TCGA dataset) and the validation set (GSE14520 and GSE76427 datasets). In the discovery set, univariate, LASSO and multivariate Cox regression analysis were employed to screen RRGs from DEGs. Using the "survival" R package, univariate analysis was used to make a primary selection of DEGs and those with $p < 0.05$ were regarded as potential candidates associated with the recurrence of HCC. Afterward, we employed the LASSO logistic regression model using the "glmnet" R package to further select genes from the potential candidates. In the LASSO-penalized regression, the optimal lambda was determined using 10-fold cross-validation and the L1 penalty was used to shrink regression coefficients toward zero. Many variables were excluded based on the principle that a larger penalty led to fewer predictive factors (*Goeman, 2010*). Consequently, candidate RRGs were considered as DEGs with nonzero coefficients. After subsampling the dataset with 1,000 iterations, the seed DEGs were shrunk into various sets and those containing DEGs with nonzero coefficients were defined as models with potential prognostic ability. DEGs with at least 900 occurrence frequencies were considered as candidate RRGs. Finally, stepwise multivariate regression analysis was conducted to identify RRGs and to construct a gene-based risk model which could predict HCC recurrence after liver resection. Using X-tile software (*Camp, Dolled-Filhart & Rimm, 2004*), the optimal cut-off points of the risk scores and the expression levels of RRGs were found. The Kaplan–Meier (KM) analysis was used to evaluate the performance of RRGs and the risk model. Furthermore, ROC analysis and calibration curves were used in validating the performance of the risk model and the "pROC" and "rms" R packages were used, respectively. In addition, the same methods were used to assess the performance of the RRGs and the risk model in the two validation datasets. In these statistic processes, a two-sided $p < 0.05$ was considered statistically significant. Finally, using "pheatmap" and "forestplot" R packages, the results were visualized by heatmap plots and forest plots, respectively.

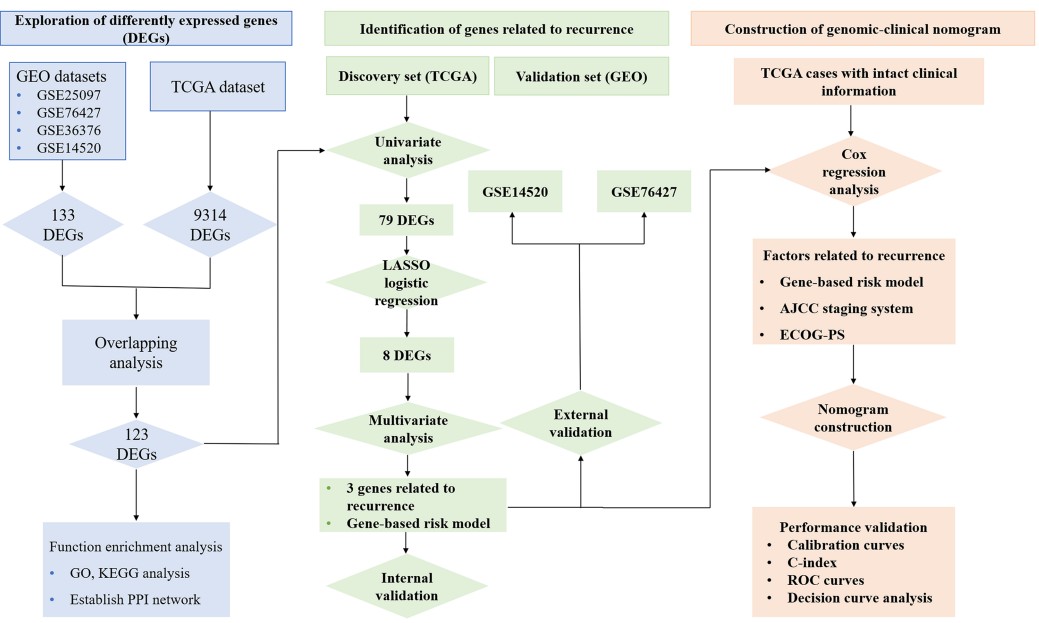

**Figure 1 Workflow of exploring RRGs and establishing genomic-clinical nomogram.** RRGs, Recurrence-related genes; GEO, The Gene Expression Omnibus; TCGA, The Cancer Genome Atlas; GO, Gene Ontology; KEGG, Kyoto Encyclopedia of Genes and Genomes; ECOG-PS, Eastern Cooperative Oncology Group performance status; ROC, receiver operating characteristic; C-index, the Harrell's concordance index.               

## Development and validation of nomogram

Univariate and multivariate regression analyses using the RNA-seq and clinical traits of the TCGA dataset combined with the gene-based risk model were used to find clinical and molecular factors independently associated with HCC recurrence. Based on the results of multivariate analysis, a genomic-clinical nomogram was established using the "rms" R package to predict 1-, 2-, and 3-year RFS for HCC patients (*Wang et al., 2013*). The performance of the nomogram was evaluated using calibration curves. The calibration was conducted with 1,000 bootstrap samples to reduce the bias. The discrimination of the nomogram was assessed using C-index and area under the curve (AUC), which measured the predictive ability of the nomogram for patients with different outcomes. The larger the C-index and a bigger AUC indicated a higher accuracy of prognostic predication (*Huitzil-Melendez et al., 2010*). Finally, using an "rmda" R package, DCA was conducted to assess the clinical usefulness of the nomogram, which could estimate the net benefits for different prediction models across all possible risk thresholds (*Vickers et al., 2008*).

# RESULTS

## Exploration of DEGs

Figure 1 reveals the workflow of this study. First, four GEO datasets, GSE25097, GSE36376, GSE14520 and GSE76427 containing 1,812 samples (1,037 HCC samples and 775 non-tumor samples) were obtained to screen DEGs and a total of 1,454, 413, 1,088 and 398 genes were found dysregulated in the tumor tissues, respectively (Figs. S1A–S1E).

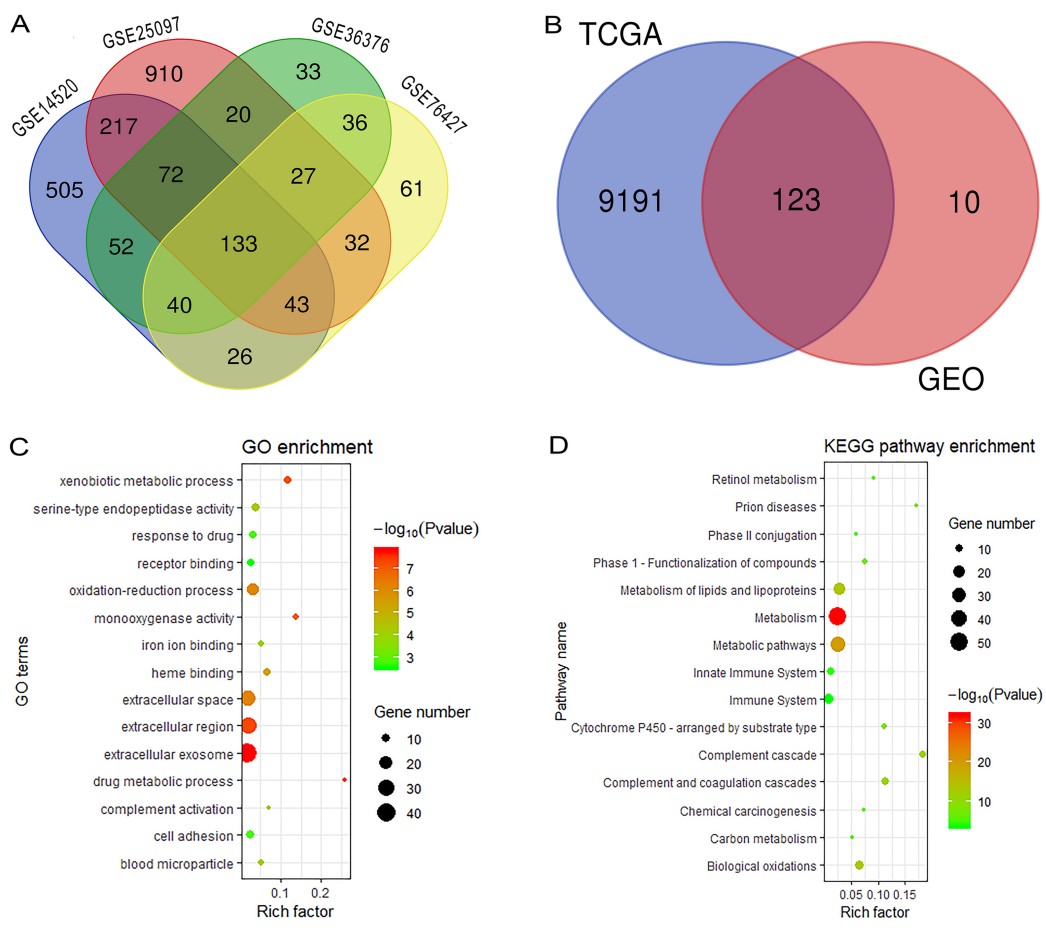

**Figure 2** **Overlapping analysis and functional enrichment analysis.** (A) Overlapping analysis for the four GEO datasets; (B) Overlapping analysis for GEO and TCGA databases; (C) GO analysis for DEGs; (D) KEGG pathways enrichment analysis for DEGs. DEGs, Differently expressed genes; GEO, the Gene Expression Omnibus; TCGA, The Cancer Genome Atlas; GO, Gene Ontology; KEGG, Kyoto Encyclopedia of Genes and Genomes.

Meanwhile, 9,314 dysregulated genes were identified in the TCGA dataset (Fig. S1F). After an overlapping analysis, 123 genes containing 12 up-regulated genes and 111 down-regulated genes were regarded as DEGs aberrantly expressed in HCC samples compared to non-tumor samples and were selected for further analysis (Figs. 2A and 2B).

## Functional enrichment analysis and construction of PPI network complex

We performed GO and KEGG analysis to elucidate the functional characteristics of the DEGs. The GO analysis indicated that the DEGs were significantly enriched in the extracellular exosome, extracellular region, extracellular space, oxidation-reduction process, cell adhesion and serine-type endopeptidase activity. In the KEGG pathways analysis, we could find that the DEGs were mainly enriched in metabolism, metabolic pathways, metabolism of lipids and lipoproteins, biological oxidations, metabolism of amino acids and derivatives, and the immune system (Figs. 2C and 2D). Finally, a PPI
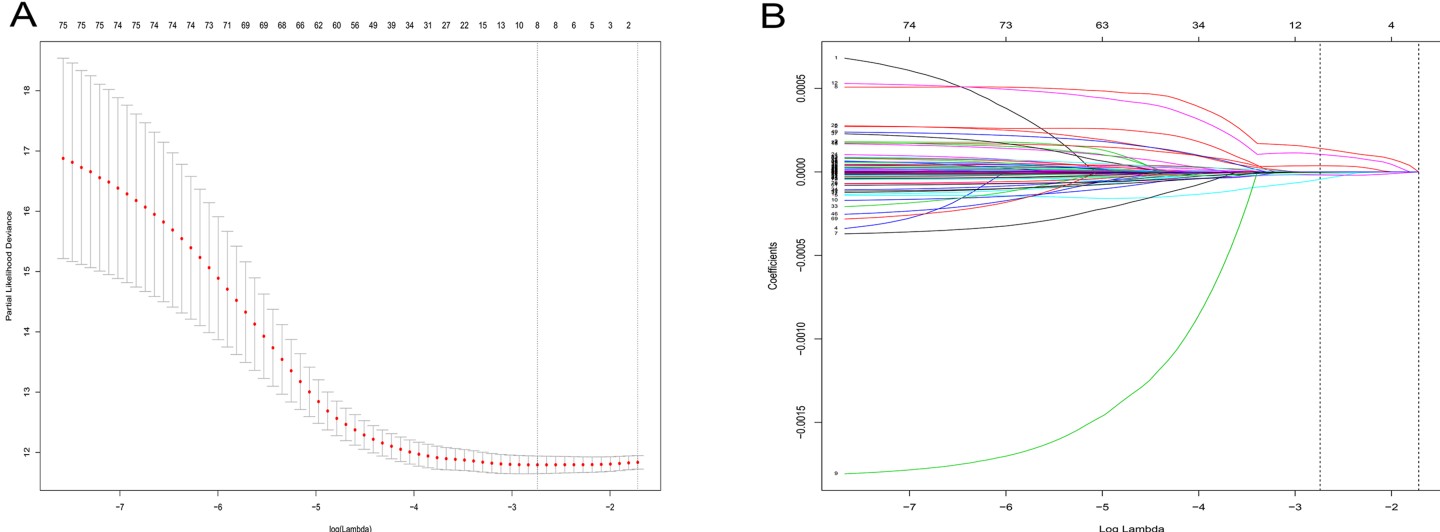

**Figure 3 LASSO logistic regression analysis for selection of RRGs from DEGs.** (A) Selection of tuning parameter (lambda) in the LASSO model via 10-fold cross-validation; (B) LASSO coefficients produced by the regression analysis (in A). RRGs, Recurrence-related genes; DEGs, differently expressed genes.               

network complex, containing 121 nodes and 773 edges, was constructed to elucidate the relationships among the DEGs (Fig. S2).

## Identification of recurrence-related genes

After a rigorous screening, 252 cases from the TCGA dataset were selected for the identification of RRGs from the discovery set. Using univariate analysis, we found that 79 DEGs were significantly related to HCC recurrence with a *p*-value < 0.05. LASSO logistic regression analysis was used to find candidates associated with recurrence and the optimal tuning parameters related to the minimum generalization error were determined from 10-fold cross-validation (Fig. 3). Consequently, eight DEGs (*PZP*, *C1RL*, *PRC1*, *PTTG1*, *UBE2C*, *AFM*, *SPP2*, *HGFAC*) were screened. Finally, we performed stepwise multivariate regression analysis and found that three genes, *PZP*, *SPP2* and *PRC1*, were biomarkers independently associated with HCC recurrence and were included in the construction of a gene-based risk model. Using X-tile software, the optimal cut-off values of the three RRGs' expression levels were found in the discovery and validation sets, and the HCC patients were divided into two groups (low-risk group and high-risk group). The KM analysis showed that the three RRGs were significantly related to HCC recurrence among all of the three datasets (Fig. 4).

According to the relative coefficient in the regression model, a risk score could be calculated to evaluate the risk of recurrence for each patient based on the expression levels of RRGs: risk score = ($-0.0793 \times$ *PZP* expression level) + ($0.2295 \times$ *PRC1* expression level) + ($-0.0662 \times$ *SPP2* expression level). We then used X-tile software to select the optimal cut-off value of the risk score for recurrence, and the patients were divided into a low-risk group or a high-risk group in the discovery and validation sets, respectively (Fig. S3).

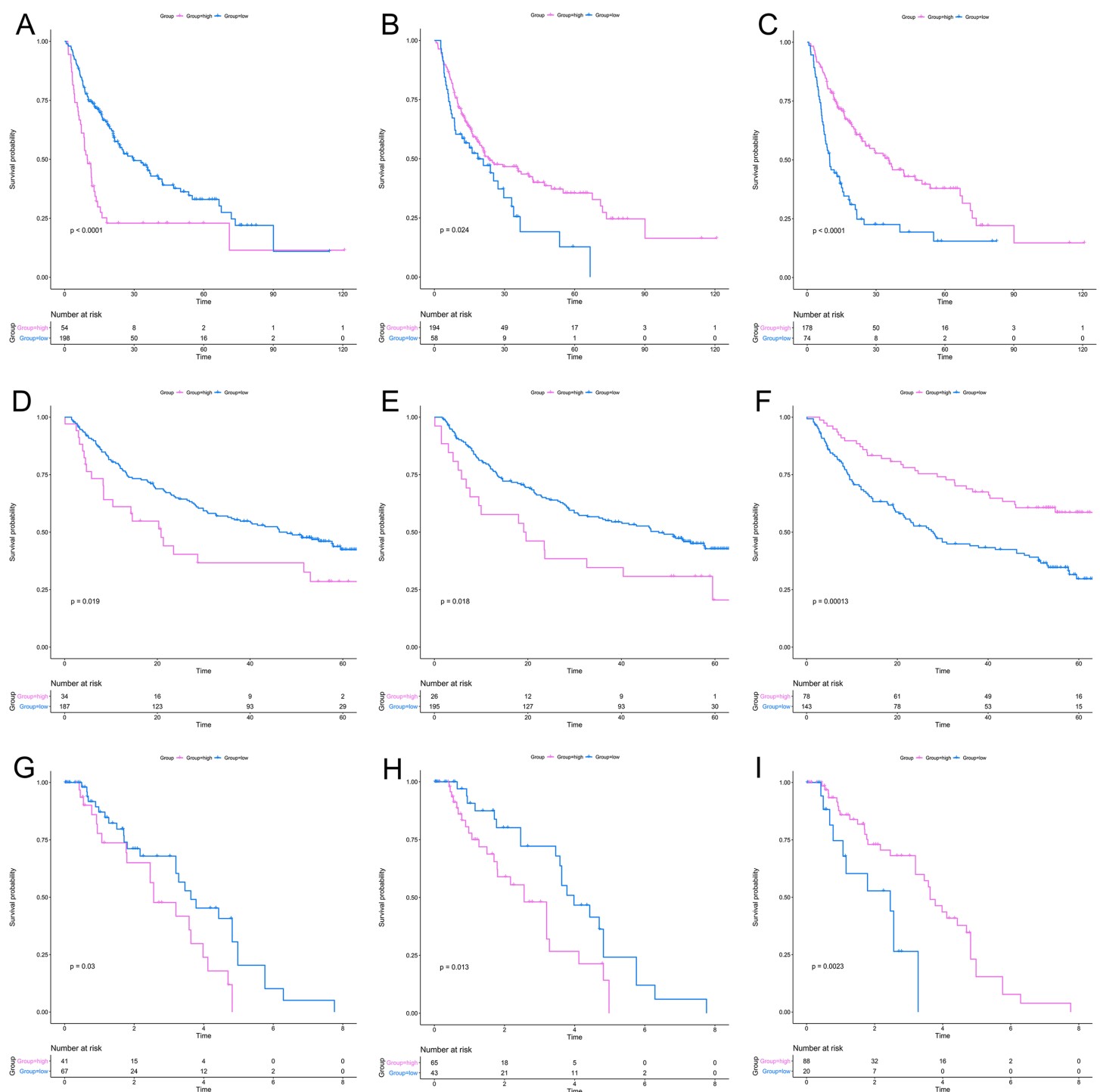

**Figure 4 KM analysis for RRGs in TCGA, GSE14520 and GSE76427 datasets.** KM analysis for PRC1 (A, D, G), PZP (B, E, H) and SPP2 (C, F, I) in TCGA (A–C), GSE14520 (D–F) and GSE76427 (G–I) datasets, respectively. KM analysis, Kaplan–Meier analysis; TCGA, The Cancer Genome Atlas.

The RFS in the low-risk group was significantly longer than that of the high-risk group among all of the three datasets (Figs. 5A–5C). The C-index of the risk model in the TCGA, GSE14520, and GSE76427 datasets was 0.663 (95% CI [0.615–0.711]), 0.571 (95% CI

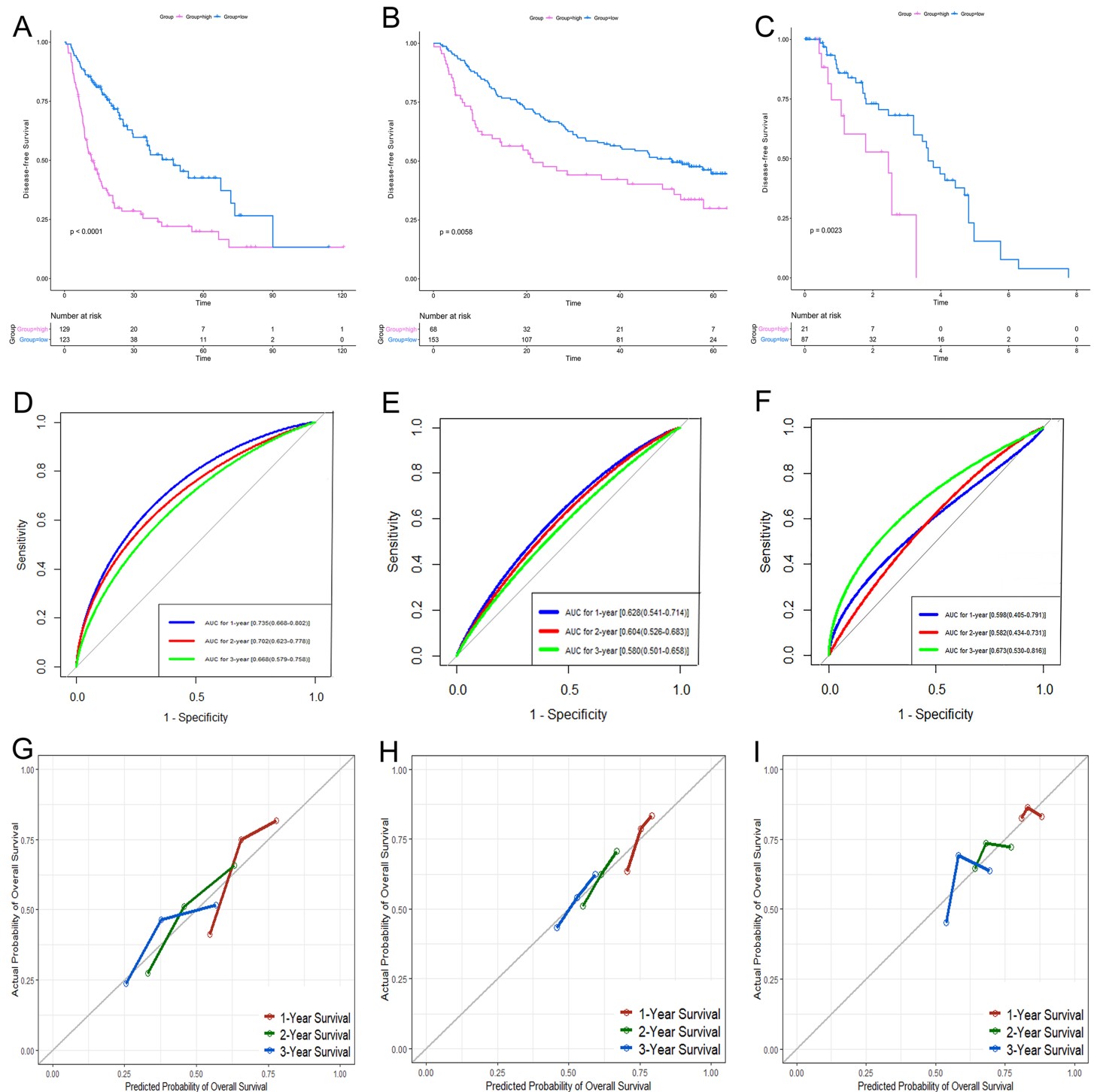

**Figure 5 Performance of the gene-based risk model in TCGA, GSE14520 and GSE76427 datasets.** KM analysis, AUC, and calibration curves for the gene-based risk model in TCGA (A, D, G), GSE14520 (B, E, H) and GSE76427 (C, F, I) datasets, respectively. KM analysis, Kaplan–Meier analysis; AUC, area under the curve; TCGA, The Cancer Genome Atlas.

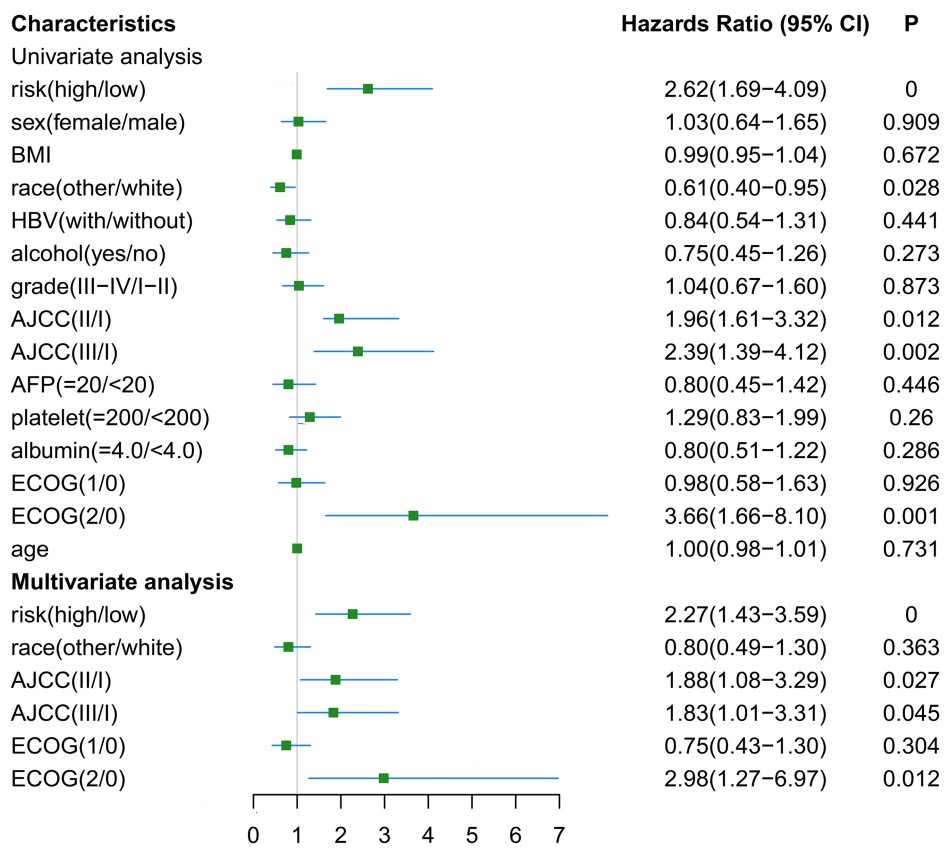

**Figure 6 Univariate and multivariate analysis for gene-based risk model and clinical characteristics with postoperative recurrence.** BMI, Body Mass Index; HBV, hepatitis B virus; AFP, alpha fetoprotein; ECOG, Eastern Cooperative Oncology Group.

[0.519–0.623]) and 0.599 (95% CI [0.492–0.706]), respectively. The ROC curves also suggested that the risk model had a good predictive efficiency among all of the three datasets (Figs. 5D–5F). Finally, the calibration curves suggested that the risk model had a good predictive efficiency compared to the observed outcomes for 1-, 2- and 3-year postoperative recurrence (Figs. 5G–5I).

## Identification of genomic and clinical factors associated with recurrence

The RNA-seq and clinical characteristics of the TCGA dataset were used for exploring genomic and clinical factors related to recurrence. After removing patients who were diagnosed with fibrolamellar carcinoma and combined hepatocellular-cholangiocarcinoma, and who did not have intact clinical information (including age, height, weight, race, hepatitis B virus status, alcohol consumption, tumor grade, AJCC stage, APF level, platelet level, albumin level, Eastern Cooperative Oncology Group performance status (ECOG-PS) and sex), a total of 162 HCC patients with both intact genomic and clinical information were included for further analysis. Univariate and multivariate cox regression analyses were used to find factors associated with HCC recurrence. As shown in Fig. 6, gene-based risk model, race, AJCC staging system and ECOG-PS were found to be potential factors related to

HCC recurrence in univariate analysis. Furthermore, the multivariate analysis indicated that the gene-based risk model, AJCC staging system and ECOG-PS were factors associated with HCC recurrence (Fig. 6).

## Construction and validation of genomic-clinical nomogram

Based on the results of the multivariate analysis, a genomic-clinical nomogram including the gene-based risk model, AJCC staging system, and ECOG-PS was established to predict the 1-, 2-, and 3-year RFS for HCC patients (Fig. 7A). The calibration curves of the nomogram for predicting the 1-, 2-, and 3-year probability of recurrence indicated that it performed well (Fig. 7B). The C-index of the nomogram was 0.678 (95% CI [0.618–0.738]).

The DCA results showed that the nomogram was more clinically useful than using the gene-based risk model, AJCC staging system, or ECOG-PS alone (Figs. 7C–7E). Afterward, we performed ROC analysis to assess the discrimination ability of the genomic-clinical nomogram and the results suggested that the AUC of the nomogram in predicting the 1-, 2-, and 3-year probability of recurrence was greater than that of using the gene-based risk model or AJCC staging system alone (Figs. 7F–7H). Finally, according to the total points of each patient in the nomogram, the patients were divided into two groups (low-risk group and high-risk group) using X-tile software, and KM analysis demonstrated that the discrimination ability of the nomogram was satisfactory ($p < 0.0001$) (Fig. 8).

## DISCUSSION

As the sixth most frequently occurring malignancy and the third leading cause of cancer-related deaths, HCC remains a significant medical problem in the world (*Forner, Reig & Bruix, 2018*). Although advances in therapeutic strategies such as liver resection, transplantation, radiofrequency ablation and transcatheter arterial chemoembolization have improved the chances of survival for certain patients, the prognosis of these patients remains unsatisfactory (*Tanwar et al., 2009*; *Zamora-Valdes, Taner & Nagorney, 2017*; *Bailey & Sydnor, 2019*; *Sun et al., 2019*). Recurrence is a significant challenge for HCC patients after liver resection and usually leading to a poor prognosis (*Chan et al., 2018*). Therefore, it is vitally important to find reliable biomarkers in the recurrent progression of the disease and to explore the mechanisms and risk factors closely associated with HCC recurrence.

In this study, we identified 123 DEGs in HCC samples compared to non-tumor samples using data obtained from GEO and TCGA databases. Univariate, LASSO, and multivariate regression analyses were used to find DEGs related to recurrence. Consequently, three RRGs were identified. A gene-based risk model was then constructed using the relative coefficients in the multivariate model and the expression levels of the three RRGs. The risk model had a good predictive efficiency for HCC recurrence. Furthermore, based on the clinical information obtained from the TCGA database, a genomic-clinical nomogram, including the gene-based risk model, AJCC staging system, and ECOG-PS was established, the C-index of the nomogram was 0.678, the calibration curves fitted well and the ROC

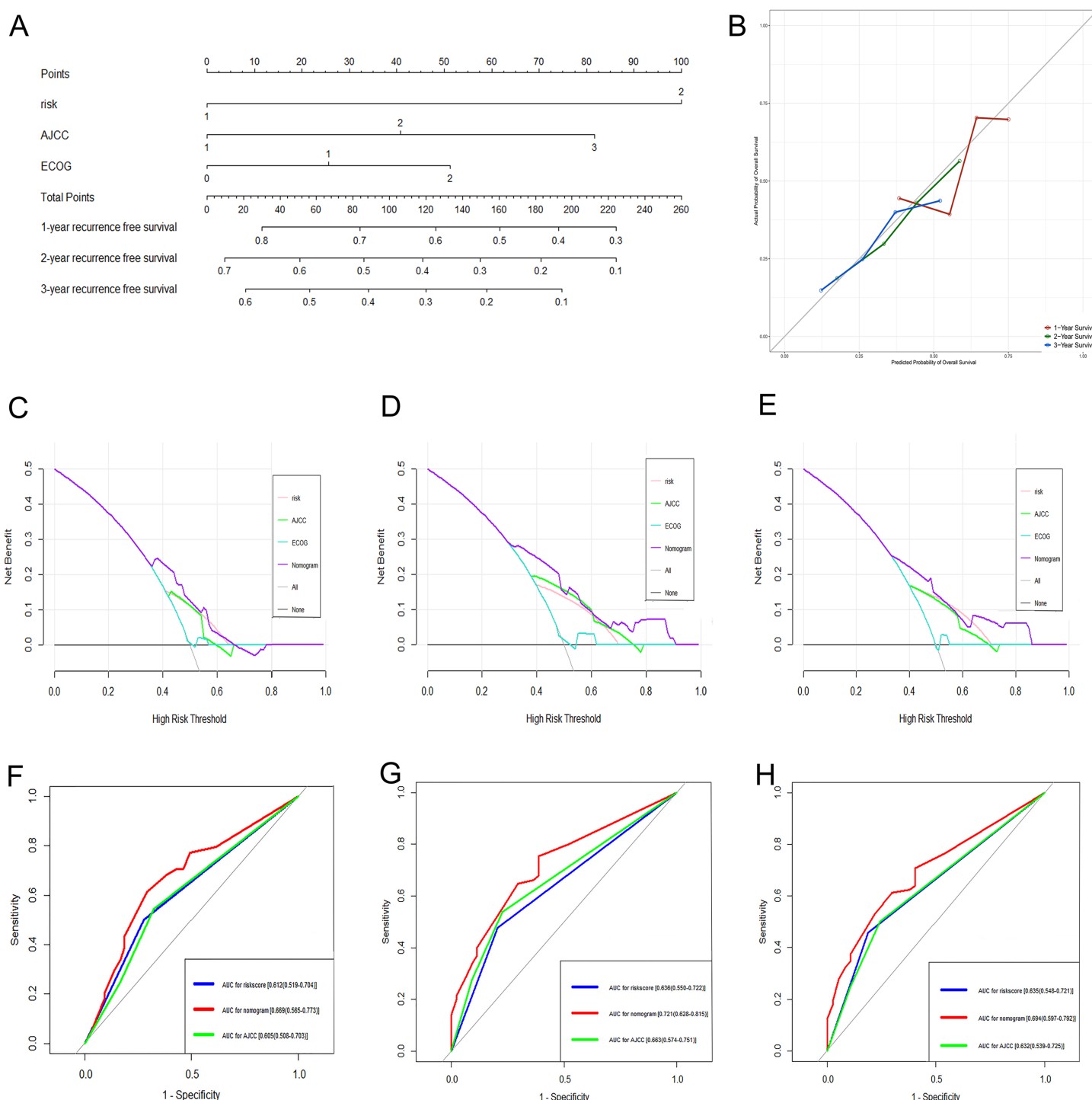

**Figure 7 Performance of the genomic-clinical nomogram in predicting postoperative recurrence in TCGA dataset.** (A) Nomogram for predicting 1-, 2- and 3-year probability of recurrence for HCC patients after liver resection; (B) Calibration curves for 1-, 2- and 3-year recurrence of the nomogram; (C–E) DCA to compare the clinical usefulness of the genomic-clinical nomogram (purple line), gene-based risk model (pink line), AJCC staging system (green line) and ECOG (turquoise line) at 1, 2 and 3 years; (F–H) ROC curves analysis to compare the predictive power of the genomic-clinical nomogram (red line), gene-based risk model (blue line) and AJCC staging system (green line) at 1, 2 and 3 years. HCC, Hepatocellular carcinoma; DCA, decision curve analysis; ECOG, Eastern Cooperative Oncology Group; ROC, receiver operating characteristic.

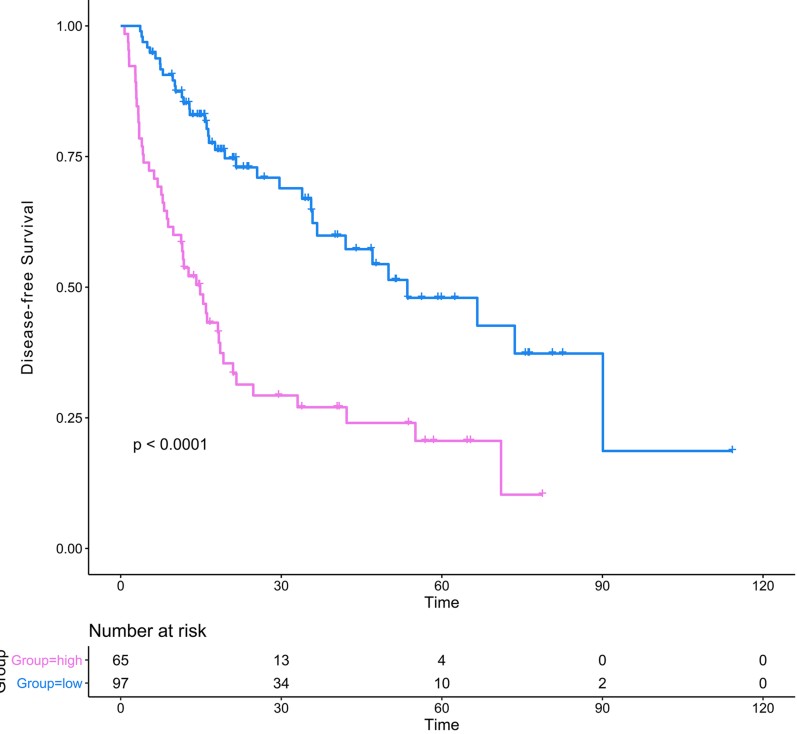

**Figure 8 Kaplan–Meier analysis of risk groups stratified using total points of the proposed nomogram.** Red lines represent patients in high risk group and blue lines represent patients in low risk group.

analysis and DCA also showed the nomogram performed well in discerning those patients with a high probability of recurrence.

Three RRGs, *PZP*, *SPP2* and *PRC1*, were identified as factors associated with HCC recurrence. *PRC1*, also known as protein regulator of cytokinesis, is a microtubule binding protein that plays crucial roles in mitosis (*Subramanian et al., 2010*). Due to its essential function in the cell cycle, dysregulation of *PRC1* could be regarded as an important factor for carcinogenesis. The development of various cancers, such as breast cancer (*Shimo et al., 2007*), lung adenocarcinoma (*Zhan et al., 2017*) and pancreatic cancer (*Nakamura et al., 2004*), was associated with the overexpression of *PRC1*. Meanwhile, the dysregulation of *PRC1* was demonstrated to be related to the early recurrence and chemoresistance of HCC (*Chen et al., 2016*; *Wang et al., 2017*). *PRC1* could contribute to the early recurrence of HCC via the regulation of the Wnt/β-catenin signaling pathway, which has been proven to be a crucial pathway in hepatocarcinogenesis and a potential therapeutic target (*Pez et al., 2013*). Furthermore, *Liu et al. (2018)* suggested that knockdown *PRC1* using siRNA could block cytokines and limit the proliferation of HCC, which could bring new insights into HCC treatment. Although previous studies have demonstrated that the other two RRGs, *PZP* and *SPP2*, were related to the prognosis of HCC, there was no study discussing their potential functions in HCC recurrence (*Yang et al., 2018*; *Zheng et al., 2018*). Robust experimental results also suggested that *SPP2* and *PZP* were associated with the stability of spliceosome and chromatin (*Warkocki et al.,*

2015; *Klein et al., 2016*), and their dysregulation might affect the progression of cell proliferation and apoptosis, which could lead to carcinogenesis and disease progression. In addition, in validation of RRGs' performance, the overexpression of *PZP* was found to be negatively associated with recurrence in the TCGA dataset, while positively related to recurrence in the GSE14520 and GSE76427 datasets. Further exploration with larger HCC and non-tumor samples should be performed to validate this issue. In summary, the mechanisms of the three RRGs in HCC development and recurrence have not been clear until now and they should be more closely followed as these could be prospective therapeutic targets for identifying the potential for HCC recurrence.

The genomic-clinical nomogram contained two clinical traits, the AJCC staging system and the ECOG-PS, aside from the gene-based risk model. The AJCC staging system is a conventional staging system used for evaluating the characteristics of cancers, selecting optimal therapeutic strategies and predicting prognosis (*Chun, Pawlik & Vauthey, 2018*). In this study, patients with AJCC stage II and III had a higher risk for HCC recurrence. According to the 8th AJCC staging system the tumors classified into stage II and III were those with vascular invasion, bigger tumor size, multiple tumor numbers, or the invasion into adjacent organs compared with the stage-I tumors. Tumors with these characteristics were more likely to have a greater invasive ability and a later tumor stage and, as a result, these patients were easier to determine as having HCC recurrence and were given higher points in the nomogram. ECOG-PS was used to assess the patients' functional capability of self-care (*Sorensen et al., 1993*). Previous studies demonstrated that ECOG-PS was closely related to the survival of cancer patients. In the BCLC staging system, ECOG-PS was listed as an independent factor for selecting therapeutic strategies and predicting prognosis (*Forner, Reig & Bruix, 2018*). Recently, a Chinese study discovered that patients with higher ECOG-PS were more likely to have poor molecular phenotypes in colorectal cancer, which could lead to a higher probability of cancer recurrence (*Chen et al., 2016*). It is clear that more focus should be placed on the functions of ECOG-PS in the recurrence of HCC, which could provide new insights into cancer treatment.

Functional enrichment analysis showed that the DEGs were mainly enriched in extracellular parts, such as the extracellular exosome, extracellular region, and extracellular space. Furthermore, metabolism-related pathways and immune-related pathways were significantly enriched by DEGs. It is well known that the liver is associated with various metabolic processes, which could help to maintain the microenvironmental stability of various tissues (*Rui, 2014*). As a result, the aberrant metabolism could provide for the unique needs of the tumor cells, contain macromolecular biosynthesis, increase energy production, and maintain the redox balance, which could provide a selective advantage for the proliferation, growth and survival of tumor cells (*De Matteis et al., 2018*). Recently, the advances of metabolomics have brought new insights into the mechanisms of HCC development, recurrence and prognosis, and the related findings could be therapeutic targets for HCC treatment (*Shang, Qu & Wang, 2016*). One of the most widely-known special mechanisms of carcinogenesis is immune invasion (*Aerts et al., 2016*). However, until the discovery of immune checkpoints, including programed cell death protein 1 (PD-1) and cytotoxic T-lymphocyte protein 4 (CTLA-4), there had not been any

breakthroughs in immunotherapy of HCC. The immune checkpoint inhibitors have been proven to possess significant clinical functions and are promising strategies for the systemic treatment of HCC (*Inarrairaegui, Melero & Sangro, 2018*).
The metabolism-related pathways and immune-related pathways played crucial roles in the recurrence of HCC and could certainly be potential therapeutic targets in the future.

Several other studies also reported the establishment of models to predict RFS for HCC after liver resection (*Cui et al., 2017*; *Gu et al., 2019*). For instance, using the gene expression file GSE76427, *Gu et al. (2019)* discovered that six long non-coding RNA (MSC-AS1, POLR2J4, EIF3J-AS1, SERHL, RMST and PVT1) were associated with a poor prognosis of HCC and constructed a nomogram including the six-lncRNA signature, TNM stage and ECOG to predict RFS for patients with small HCC. The C-index of their nomogram was 0.684 (95%CI [0.635–0.733]), which was comparable with our nomogram (C-index 0.678; 95% CI [0.618–0.738]). Furthermore, besides the gene-based signature, TNM stage and ECOG were also integrated into their nomogram, which was robust evidence for our nomogram (including the AJCC staging system and ECOG). However, Gu's study only analyzed one gene expression file with a relatively small number of samples, which could make the results inaccurate to some extent. Our study analyzed a total of 1,645 samples from four GEO datasets, containing 870 HCC samples and 775 normal samples. Consequently, the RRGs found in our study were more reliable. Combined with rigorous validations, it was clear that our nomogram was could performed well in the prediction of RFS for HCC patients after liver resection.

However, limitations existed in the current study. First of all, the expression of RRGs were only explored and validated among online databases; no experiments were conducted to confirm them. Consequently, experiments about the dysregulation of the RRGs and their mechanisms in HCC recurrence should be explored in the future. Secondly, we failed to make a comparison of the genomic-clinical nomogram to other widely accepted staging systems, such as the BCLC staging system. Lastly, we did not perform an external validation for the genomic-clinical nomogram, and further validations with additional data should be performed to validate the performance of our nomogram.

## CONCLUSIONS

Three new biomarkers were found that were related to postoperative recurrence for HCC patients after hepatectomy and a reliable genomic-clinical nomogram was established to predict the 1-, 2-, and 3-year RFS. The C-index, ROC analysis and DCA showed that the nomogram performed well. These findings could bring new insights into molecular mechanisms for HCC recurrence, contribute to HCC treatment, and help surgeons to predict the likelihood of recurrence and the prognosis for HCC patients postoperatively.

### Funding

This study was funded by the National Natural Science Foundation of China (No. 81770566) and the Department of Science and Technology of Sichuan Province

(No. 19ZDYF1682). The funders had no role in study design, data collection and analysis, decision to publish, or preparation of the manuscript.

## Grant Disclosures
The following grant information was disclosed by the authors:
National Natural Science Foundation of China: 81770566.
Department of Science and Technology of Sichuan Province: 19ZDYF1682.

## Competing Interests
The authors declare that they have no competing interests.

## Author Contributions
- Junjie Kong conceived and designed the experiments, contributed reagents/materials/analysis tools, prepared figures and/or tables, authored or reviewed drafts of the paper, approved the final draft.
- Tao Wang analyzed the data, contributed reagents/materials/analysis tools, prepared figures and/or tables, authored or reviewed drafts of the paper, approved the final draft.
- Shu Shen conceived and designed the experiments, analyzed the data, authored or reviewed drafts of the paper, approved the final draft.
- Zifei Zhang analyzed the data, authored or reviewed drafts of the paper, approved the final draft.
- Xianwei Yang analyzed the data, contributed reagents/materials/analysis tools, prepared figures and/or tables, approved the final draft.
- Wentao Wang conceived and designed the experiments, authored or reviewed drafts of the paper, approved the final draft.

## Data Availability
The raw measurements are available in the Supplementary Files.

## Supplemental Information
Supplemental information for this article can be found online at http://dx.doi.org/10.7717/peerj.7942#supplemental-information.

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
