# Peer review of "A genomic-clinical nomogram predicting recurrence-free survival for patients diagnosed with hepatocellular carcinoma"

_PeerJ, doi:10.7717/peerj.7942_

## Round 0.1 · original submission · Minor Revisions

· Academic Editor

Minor Revisions

Please address all the critiques of both reviewers and amend your manuscript accordingly.

Reviewer 1 ·

Basic reporting

The authors have elaborated the ways to explore the recurrence related genes (RRGs) and establish a genomic-clinical nomogram for predicting the postoperative recurrence of hepatocellular carcinoma. The study is well-designed, and the authors have performed the necessary controls and tests to validate their models. The manuscript is well-written with overall proper use of language and grammar and generally complies with PeerJ standards. Introduction is adequate with necessary background information needed to understand the gravity of the question being asked. The necessary raw data has been provided. The data provided corroborates the conclusions made by the authors.

Experimental design

The research though not entirely novel is critical in terms of outcome and effect on the patient diagnosis. The questions being asked about the lack of robust prediction tools for HCC recurrence up to 3 years post-operation and the nomogram constructed based on the genomic and clinical analyses is critical. All the methods have been described adequately. The checks and balances performed to validate the findings and the models are rigorous. The authors, however, should consider elaborating the significance of the tests performed i.e. the C-index, ROC and decision curve analyses (DCA).

Validity of the findings

As mentioned, the data presented supports the conclusions made by the authors. The authors have shown that the tests performed by the authors to check the performance of the nomogram, substantiate their model. However, the reviewer suggests that the authors should present a comparison of their nomogram with already established nomograms for HCC or even other cancers to further bolster their model. Other than this minor suggestion, the manuscript, overall, is suited for publication in PeerJ.

Reviewer 2 ·

Basic reporting

In this work, the authors used bioinformatics tools to analyze the gene expression profiles from GEO and TCGA databases and identified three potential candidate recurrence-related genes in hepatocellular carcinoma.

Overall, the manuscript is well-written and is self-contained except at few places as elaborated in general comments.

Experimental design

The aim of the study is well-defined and the methods used are appropriate.

Validity of the findings

Conclusions are well stated. Overall results presented in the manuscript support conclusion except at places elaborated in general comments.

Additional comments

1) On lines 158-161, it is not clear how lasso regression was used resulting in identification of 8 DEGs from 79 candidate DEGs from univariate analysis. Authors should describe this in more detail. Also, on line 161, “Consequently, 8 DEGs were screened (Fig. 3)”, figure 3 is wrongly referenced here as there is no information about 8 DEGs in this figure. Authors should mention what these eight DEGs are.
2) In the methods section, I would recommend removing the “Statistical analysis” sub-section and move the necessary information (i.e. the packages used and the criteria for different analysis) under corresponding sub-sections. For example, lines 124-127 describing information related to DEGs identification should be moved under the heading “Exploration of differentially expressed genes (DEGs) and bioinformatics analysis”.
3) Figures are wrongly referenced at several places in the manuscript. To mention a few: on line 173, “group in the discovery and validation sets, respectively (Fig. S3)”, the referenced figure should be Fig. S4. On line 174, “…..high-risk group among all of the 3 datasets (Fig. 5A-C). ”, the referenced figure should be Fig. 4A-C.

---

## Round 0.2 · accepted · Accept

· Academic Editor

Accept

All critiques were adequately addressed and the manuscript was revised accordingly. Therefore, the amended version is acceptable.

Reviewer 1 ·

Basic reporting

The authors have satisfactorily addressed the suggestions raised by this reviewer in their rebuttal and recommends that the manuscript be accepted for publication.

Experimental design

The issues stated with respect to experimental design have been addressed in the rebuttal

Validity of the findings

The issues stated previously have been addressed in the rebuttal

Reviewer 2 ·

Basic reporting

NA

Experimental design

NA

Validity of the findings

NA

Additional comments

Authors have addressed all my concerns and the manuscript is now suitable for publication.